COMMUNICATION

# Disease-associated *KCNMA1* variants decrease circadian clock robustness in channelopathy mouse models

Ria L. Dinsdale[1] , Cooper E. Roache[1] , and Andrea L. Meredith[1]

**KCNMA1 encodes the voltage- and calcium-activated K+ (BK) channel, which regulates suprachiasmatic nucleus (SCN) neuronal firing and circadian behavioral rhythms. Gain-of-function (GOF) and loss-of-function (LOF) alterations in BK channel activity disrupt circadian behavior, but the effect of human disease-associated *KCNMA1* channelopathy variants has not been studied on clock function. Here, we assess circadian behavior in two GOF and one LOF mouse lines. Heterozygous *Kcnma1*[N999S/WT] and homozygous *Kcnma1*[D434G/D434G] mice are validated as GOF models of paroxysmal dyskinesia (PNKD3), but whether circadian rhythm is affected in this hypokinetic locomotor disorder is unknown. Conversely, homozygous LOF *Kcnma1*[H444Q/H444Q] mice do not demonstrate PNKD3. We assessed circadian behavior by locomotor wheel running activity. All three mouse models were rhythmic, but *Kcnma1*[N999S/WT] and *Kcnma1*[D434G/D434G] showed reduced circadian amplitude and decreased wheel activity, corroborating prior studies focused on acute motor coordination. In addition, *Kcnma1*[D434G/D434G] mice had a small decrease in period. However, the phase-shifting sensitivity for both GOF mouse lines was abnormal. Both *Kcnma1*[N999S/WT] and *Kcnma1*[D434G/D434G] mice displayed increased responses to light pulses and took fewer days to re-entrain to a new light:dark cycle. In contrast, the LOF *Kcnma1*[H444Q/H444Q] mice showed no difference in any of the circadian parameters tested. The enhanced sensitivity to phase-shifting stimuli in *Kcnma1*[N999S/WT] and *Kcnma1*[D434G/D434G] mice was similar to other *Kcnma1* GOF mice. Together with previous studies, these results suggest that increasing BK channel activity decreases circadian clock robustness, without rhythm ablation.**

## Introduction

Voltage- and calcium-activated K+ (BK) channels are major regulators of neuronal activity (Contet et al., 2016; Latorre et al., 2017). BK channel current has been shown to regulate spontaneous firing in the suprachiasmatic nucleus (SCN) in a diurnal manner (Meredith et al, 2006; Pitts et al, 2006; Montgomery et al, 2013; Whitt et al, 2016). In the SCN, expression of the gene encoding the BK channel (*KCNMA1*) varies over the circadian cycle, with lower expression during the day and higher expression during the night (Meredith et al, 2006; Pitts et al, 2006; Kent and Meredith, 2008; Montgomery et al, 2013; Whitt et al, 2016). This pattern sets their influence over the decrease in spontaneous firing at night while facilitating spontaneous firing during the day. Supporting the role of the BK channel in SCN, there are three previous genetic alterations in *KCNMA1* that can be classified as gain-of-function (GOF) and loss-of-function (LOF). These mouse lines show that perturbations in BK channel activity in both directions result in disruptions of neural activity and

behavioral rhythms (Meredith et al., 2006; Montgomery et al., 2013; Whitt et al., 2016).

Transgenic model *Tg-BK*[R207Q], where the GOF mutation R207Q is induced by Per-1 promoter, increases BK channel current during the day and therefore decreases SCN excitability (Montgomery and Meredith, 2012; Montgomery et al., 2013). Disruption of cyclic BK current in *Tg-BK*[R207Q] mice leads to a reduction or loss of SCN rhythmicity, demonstrated by multielectrode array recordings (Montgomery et al., 2013). The reduction in SCN rhythmicity did not completely abolish behavioral rhythmicity. However, *Tg-BK*[R207Q] mice demonstrate an elongated active period (α) and an increased sensitivity to phase-shifting stimuli (Montgomery et al., 2013).

A second GOF model previously studied for the impact on circadian rhythms is the knockout of the *KCNMB2* accessory subunit (β2; Whitt et al., 2016). The β2 subunit causes inactivation of the BK channel current, and removal of inactivation causes GOF BK channel behavior (Whitt et al., 2016).

[1]Department of Physiology, University of Maryland School of Medicine, Baltimore, MD, USA.

Correspondence to Andrea L. Meredith: ameredith@som.umaryland.edu

This work is part of a special issue on Structure and Function of Ion Channels in Native Cells and Macromolecular Complexes.

Like $Tg\text{-}BK^{R207Q}$, the loss of β2 increases BK current during the day, producing a decrease in the daytime firing rate to the same levels as nighttime and a loss of rhythmicity in SCN multielectrode array recordings (Whitt et al., 2016). β2 KO had a similarly altered response to phase-shifting stimuli as $Tg\text{-}BK^{R207Q}$ and showed a reduction in circadian behavioral amplitude caused by an increase in activity during subjective night (Whitt et al., 2016).

In contrast, the LOF BK channel-null model, $Kcnma1^{-/-}$ mice, also showed disrupted SCN firing and rhythmicity, together with a primary circadian behavioral deficit (Meredith et al., 2006; Kent and Meredith, 2008). $Kcnma1^{-/-}$ SCN neurons increase spontaneous firing rate during subjective night by removing the BK channel current (Meredith et al., 2006), reducing SCN rhythmicity by multielectrode array recordings (Meredith et al, 2006; Kent and Meredith, 2008). The SCN-level changes significantly impact the circadian behavior of the animal, where 10% of $Kcnma1^{-/-}$ mice are completely arrhythmic. The $Kcnma1^{-/-}$ mice that remain rhythmic have a major decrease in circadian amplitude, increase in circadian period, and an increased sensitivity to phase-shifting stimuli (Meredith et al., 2006).

These mouse lines comprise engineered GOF and LOF mutations manipulating BK channel gating, but recently a new human disorder resulting from mutations in $KCNMA1$ has been characterized (Bailey et al., 2019; Miller et al., 2021; Park et al., 2022). The comprehensive impact of patient mutations on various neurological functions is in the beginning stages of being studied. Although some $KCNMA1$ channelopathy patients anecdotally report sleep disturbances, it is difficult to disentangle an antecedent circadian defect from other neurological-associated symptoms in this new disorder. Therefore, our study aimed to directly characterize any circadian rhythm deficits associated with three of these variants in mouse models under laboratory-controlled conditions.

In this study, we evaluate the circadian rhythm of three recently generated mice models of human $KCNMA1$ channelopathy variants, $Kcnma1^{N999S/WT}$, $Kcnma1^{D434G/D434G}$, and $Kcnma1^{H444Q/H444Q}$ (Park et al., 2022). The two GOF $KCNMA1$ variants, N999S and D434G, make up about a third of the patient population (Miller et al., 2021). Both variants cause a shift in the voltage-dependence of activation to more negative potentials, faster activation, and slower deactivation (Du et al, 2005; Díez-Sampedro et al, 2006; Wang et al, 2009; Berkefeld and Fakler, 2013; Li et al, 2018; Plante et al, 2019; Moldenhauer et al, 2020; Dong et al, 2022; Park et al, 2022). The LOF H444Q variant causes a shift in the voltage-dependence of activation to more positive potentials, demonstrating slower activation and faster deactivation (Park et al., 2022). Patients and mice harboring GOF variants N999S and D434G experience the episodic hypokinetic motor disorder paroxysmal non-kinesigenic dyskinesia-3 (PNKD3), while mice harboring LOF variant H444Q have a distinct hyperkinetic motor phenotype (Park et al., 2022). In the setting of these different types of dyskinesias, we sought to assess circadian behavior by analyzing locomotor wheel-running activity and neuronal firing patterns in the SCN by whole-cell electrophysiology.

## Materials and methods

### Mice

$Kcnma1^{N999S/WT}$, $Kcnma1^{D434G/D434G}$, and $Kcnma1^{H444Q/H444Q}$ mice were created and backcrossed on a C57BL/6J background as described previously (Park et al., 2022). Transgenic and littermate controls were produced from heterozygous $Kcnma1^{N999S/WT}$ male and C57BL/6J female breeders, or heterozygous $Kcnma1^{D434G/WT}$ and $Kcnma1^{H444Q/WT}$ breeding pairs. Genotyping was performed from tail snips by TaqMan real-time PCR at Transnetyx, Inc, as described previously (Park et al., 2022).

For all studies, mice were split by sex and group-housed on a standard 12:12 h light:dark (LD) cycle until experimental procedures. Time points over the circadian cycle are referred to as zeitgeber time (ZT), denoting time in hours relative to the 24-h cycle, with lights on being defined as ZT0, and lights off at ZT12. All procedures involving mice were approved by the University of Maryland School of Medicine Institutional Animal Care and Use Committee.

### Circadian behavioral rhythm

For locomotor rhythms, mice (2–4 mo old) were housed individually in cages containing a running wheel for at least 10 d in 12:12-h LD and 15 d in constant darkness (DD). The activity was sampled every 1 min in ClockLab software (Actimetrics). Actograms were constructed by double-plotting consecutive days of activity over the recording period. Circadian period (τ), χ² amplitude, and Fast Fourier transform (FFT) relative power (rPSD) for 0.04–0.042 cycles/h were determined from the last 10 d in LD and the first 15 d of wheel running activity in DD in ClockLab software. Activity period (α) was determined as the length of time an animal had consolidated activity using default settings with manual adjustment, with ρ defined as the portion of the cycle outside of α. For re-entrainment experiments, after 7 d of stable entrainment, the LD cycle was phase advanced by 6 h. The response was calculated as the number of days to stabilize re-entrainment of at least 3 d. After keeping mice in DD for 7 d, phase shifts in response to a light pulse were calculated as the number of hours between the activity onset regression fits the day before and after a 30 min light pulse delivered in early subjective night (CT16). All bout parameters (bout duration, bouts/day, counts/bout) were calculated from the first 15 d in DD, where a bout of activity is defined as periods during which the activity does not fall below 30 counts/min for >10 min. Data was excluded from mice that failed to record activity counts for two consecutive days. All animal experiments were conducted blinded to genotype during data collection and analysis from at least three independent cohorts for each transgenic line. Data were obtained from males and females, and no randomization was used to allocate animals into cohorts. WT controls were compared with transgenic littermates within each individual transgenic line. For all figures, the WT control actogram is a WT mouse from a $Kcnma1^{N999S/WT}$ × C57BL/6J litter.

### Acute SCN slice preparation and electrophysiological recordings

Brains were harvested during the light portion of the circadian cycle at ZT 0–4 from 3- to 6-wk-old mice. Brains were rapidly

placed into ice-cold sucrose-substituted saline containing the following (in mM): 5 $MgCl_2$, 26 $NaHCO_3$, 1.25 $Na_2HPO_4$, 3.5 KCl, 0.05 $CaCl_2$, 10 glucose, 200 sucrose, 1.2 sodium pyruvate, and 0.4 vitamin C, aerated with 95% $O_2$ and 5% $CO_2$. Acute coronal slices were cut at 300 μm on a VT1000S vibratome (Leica Microsystems) at 3–4°C. Slices containing SCN were recovered for 1–2 h at 25°C submerged in oxygenated artificial cerebrospinal fluid (in mM: 125 NaCl, 1.7 $MgCl_2$, 26 $NaHCO_3$, 1.25 $Na_2HPO_4$, 3.5 KCl, 2 $CaCl_2$, 10 glucose, 1.2 sodium pyruvate, and 0.4 vitamin C) in a recovery chamber (BSK-AM; Sci Sys). Acute slices were transferred to the recording chamber (RC26GLP/PM-1; Warner Instruments) with gravity flow bath perfusion of 1–2 ml/min oxygenated artificial cerebrospinal fluid at RT. Neurons were visualized with a Luca-R DL-604 EMCCD camera (Andor) under IR-DIC illumination on an FN1 upright microscope (Nikon). Current-clamp recordings were made with a Multiclamp 700B amplifier and pCLAMP 11.2 software (Molecular Devices). Data were acquired at a 50-kHz sampling rate. The recording window was at the peak (ZT4–8) of the circadian rhythm. All recordings were made with synaptic transmission intact to more closely approximate in vivo activity. Electrodes (4–8 MΩ resistance) were filled internal solution (in mM: 123 K-methane-sulfonate, 9 NaCl, 0.9 EGTA, 9 HEPES, 14 Tris-phosphocreatine, 2 Mg-ATP, 0.3 Tris-GTP, and 2 $Na_2$-ATP, pH adjusted to 7.3 with KOH). After gigaseal formation and whole cell break-in, membrane properties were elicited from a +20-mV voltage step from a holding potential ($V_h$) of –70 mV. Access resistance was verified to be <30 MΩ with <20% change at the end of the recording. Series resistance was compensated at 60–70%. For action potentials, data were acquired in 10-s sweeps. Silent cells were identified by injecting a 20-pA current to elicit an action potential. Frequency was calculated as the average of each sweep. Baseline potential was determined as the average interspike potential in the presence of spontaneous activity. Input resistance was calculated as the linear slope of the voltage from hyperpolarizing current injections (–10 to –25 pA in 5 pA increments). Instantaneous frequency histograms were constructed by normalizing the number of events in 0.5-Hz bins to the total number of events from all neurons (GraphPad Prism v9.3.0). All data were corrected for liquid junctional potential (10 mV).

## Statistics

Electrophysiology and behavior data were tested for normality with Shapiro–Wilk normality test and either parametric or non-parametric statistic tests were analyzed in GraphPad Prism v9.3.0. For the parametric test, two-tailed unpaired $t$ tests were performed with Welch's correction for unequal variance. Mann–Whitney test was used for non-parametric data. Data are plotted as individual data points with median and interquartile range to evaluate data variability or as mean ± SEM, as indicated in the figure legends. P < 0.05 was considered significant.

## Online supplemental material

Table S1 contains a summary of behavioral data and SCN neuronal recordings of data presented in Figs. 1, 2, 3, and 4.

## Results

### Evaluation of primary circadian behavior in *Kcnma1*[N999S/WT], *Kcnma1*[D434G/D434G], and *Kcnma1*[H444Q/H444Q] mice

In this study, we assessed basic circadian rhythm from the periodicity and amplitude of locomotor wheel-running activity. Mice were first housed in a standard 12:12-h LD cycle to test the effect of standard light cycle entrainment on the circadian pacemaker. All lines entrained normally to the standard 12:12-h LD cycle, demonstrated in running wheel actograms and behavioral analyses collected in LD conditions (Fig. 1 A and Table S1).

To assess the intrinsic circadian rhythm in the absence of light, mice were housed in DD (Fig. 1 A). The circadian period was assessed using $χ^2$ statistical analysis (Fig. 1 B). Based on changes in circadian behavior observed with previously engineered GOF and LOF *Kcnma1* mutations (Meredith et al, 2006; Kent and Meredith, 2008; Montgomery et al, 2013; Whitt et al, 2016), we predicted that the human patient variants would cause a disruption of the circadian behavior of *Kcnma1*[N999S/WT], *Kcnma1*[D434G/D434G], and *Kcnma1*[H444Q/H444Q] mice. For the GOF lines in DD, we found a small change in the circadian period. The circadian period (τ) for *Kcnma1*[D434G/D434G] mice was 13 min shorter compared with WT littermates, while *Kcnma1*[N999S/WT] trended toward a reduction in the average period that was not significant (Fig. 1 C and Table S1). Testing the LOF line, *Kcnma1*[H444Q/H444Q] mice showed no difference in the circadian period compared with WT littermates (Fig. 1 C and Table S1).

To assess circadian behavioral amplitude in DD, the circadian peak from $χ^2$ periodogram analysis and FFT were quantified. Both tests assess the robustness of the circadian rhythm from the amplitude of the circadian signal using the peak of the $χ^2$ periodogram signal and the relative power of the circadian frequency band (0.04–0.042 cycles per hour), respectively (Sokolove and Bushell, 1978). For both *Kcnma1*[N999S/WT] and *Kcnma1*[D434G/D434G] lines, there was a reduction in both the $χ^2$ circadian amplitude and FFT relative power of the circadian rhythm (Fig. 1, C and D; and Table S1). In the case of *Kcnma1*[H444Q/H444Q] mice, there was no change in circadian amplitude (Fig. 1, D and E; and Table S1). To test if any of the transgenic mice models had a change in the active period, the average time of consolidated running activity (α) was assessed. However, neither of the GOF mice models *Kcnma1*[N999S/WT] or *Kcnma1*[D434G/D434G] showed any difference in α length nor did the LOF variant, *Kcnma1*[H444Q/H444Q] (Fig. 1 F and Table S1). Based on the reduction in period and circadian amplitude, there is evidence to suggest the potential for a primary circadian rhythm deficit in *Kcnma1*[D434G/D434G], with a more limited potential in *Kcnma1*[N999S/WT]. There is no evidence of a primary circadian deficit in *Kcnma1*[H444Q/H444Q] mice.

*Kcnma1*[N999S/WT] and *Kcnma1*[D434G/D434G] are models for paroxysmal non-kinesigenic dyskinesia-3 (PNKD3) and demonstrate bouts of immobility (Park et al., 2022; Dong et al., 2022). This could affect the ability of the mice to run on the wheels and therefore impact the measurement of circadian amplitude due to $χ^2$ analysis being sensitive to wheel rotations. To test this, the baseline ability of the mice to run on the wheels was assessed by the number of wheel counts for each line. Total wheel counts for

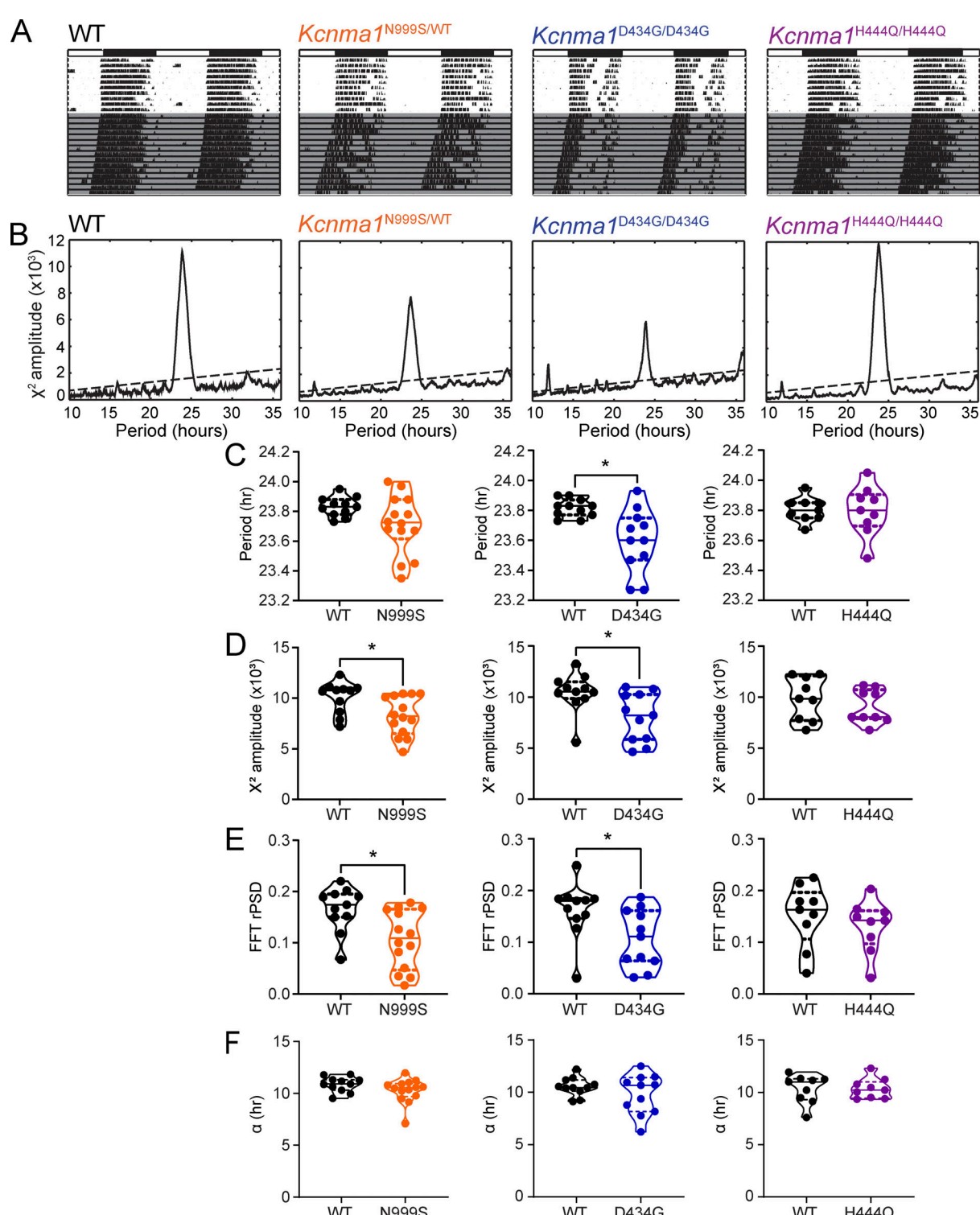

Figure 1. **Evaluation of circadian rhythm from wheel running behavior. (A)** Representative double-plotted running wheel actograms for WT, *Kcnma1*[N999S/WT], *Kcnma1*[D434G/D434G], and *Kcnma1*[H444Q/H444Q] mice. White and black bars denote 12:12-h LD cycle for first 10 d and the grey shaded area denotes DD (last 15 d). **(B)** $\chi^2$ periodograms for WT, *Kcnma1*[N999S/WT], *Kcnma1*[D434G/D434G], and *Kcnma1*[H444Q/H444Q] mice, generated from DD activity data in A. The dotted line denotes a 0.01 confidence interval. **(C–F)** Average circadian period (C; τ; WT versus *Kcnma1*[D434G/D434G], P = 0.0068, *t* test, n = 11, 11, respectively), (D) $\chi^2$ amplitude (WT versus *Kcnma1*[N999S/WT], P = 0.0163, *t* test, n = 11, 14, respectively; WT versus *Kcnma1*[D434G/D434G], P = 0.0192, MWU, n = 11, 11, respectively), (E) the relative power of the dominant circadian component of a Fourier transform (WT versus *Kcnma1*[N999S/WT], P = 0.0073, *t* test; WT versus *Kcnma1*[D434G/D434G], P = 0.0352, MWU, n = 11, 11, respectively), and (F) average active period (α) of DD activity data from A for WT, *Kcnma1*[N999S/WT], *Kcnma1*[D434G/D434G], and *Kcnma1*[H444Q/H444Q] mice as indicated. Data is presented as individual data points with median and interquartile range.

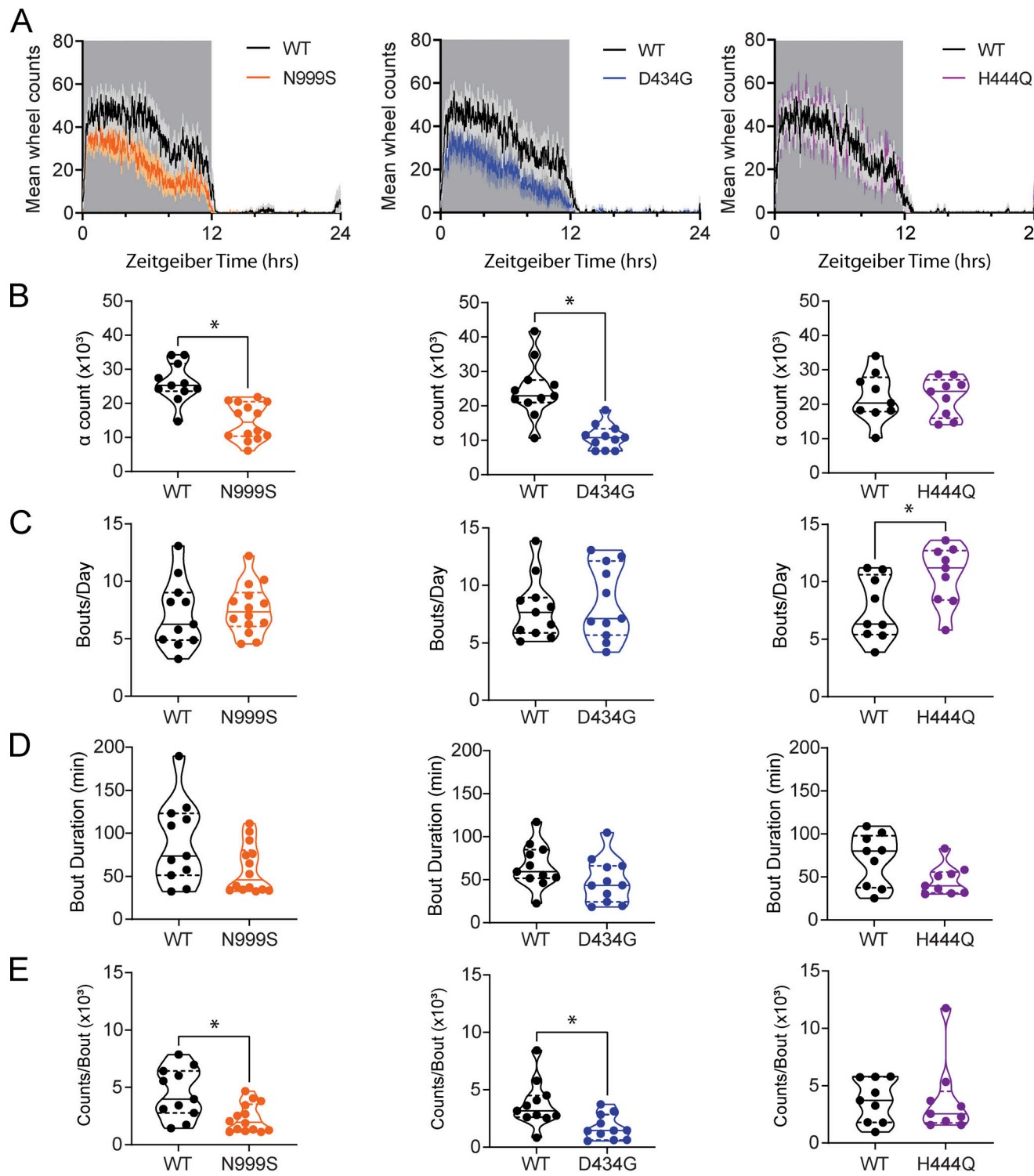

Figure 2. **Analysis of locomotor activity bouts. (A)** Activity profiles of average counts per minute for *Kcnma1*[N999S/WT], *Kcnma1*[D434G/D434G], and *Kcnma1*[H444Q/H444Q] mice and WT littermates as indicated. Grey shaded area denotes times of lights off. **(B–E)** Average number of counts during consolidated subjective night activity (B; α counts; WT versus *Kcnma1*[N999S/WT], P < 0.0001, *t* test, *n* = 11, 14, respectively; WT versus *Kcnma1*[D434G/D434G], P = 0.0002, *t* test, *n* = 11, 11, respectively), (C) average number of bouts per day (WT versus *Kcnma1*[H444Q/H444Q], P = 0.0292, *t* test, *n* = 9, 9, respectively), (D) average duration of bouts, and (E) average number of counts per bout (WT versus *Kcnma1*[N999S/WT], P = 0.0090, *t* test, *n* = 11, 14, respectively; WT versus *Kcnma1*[D434G/D434G], P = 0.0105, *t* test, *n* = 11, 11, respectively), for WT, *Kcnma1*[N999S/WT], *Kcnma1*[D434G/D434G], and *Kcnma1*[H444Q/H444Q] mice housed in DD, as indicated. Data are presented as individual data points with median and interquartile range.

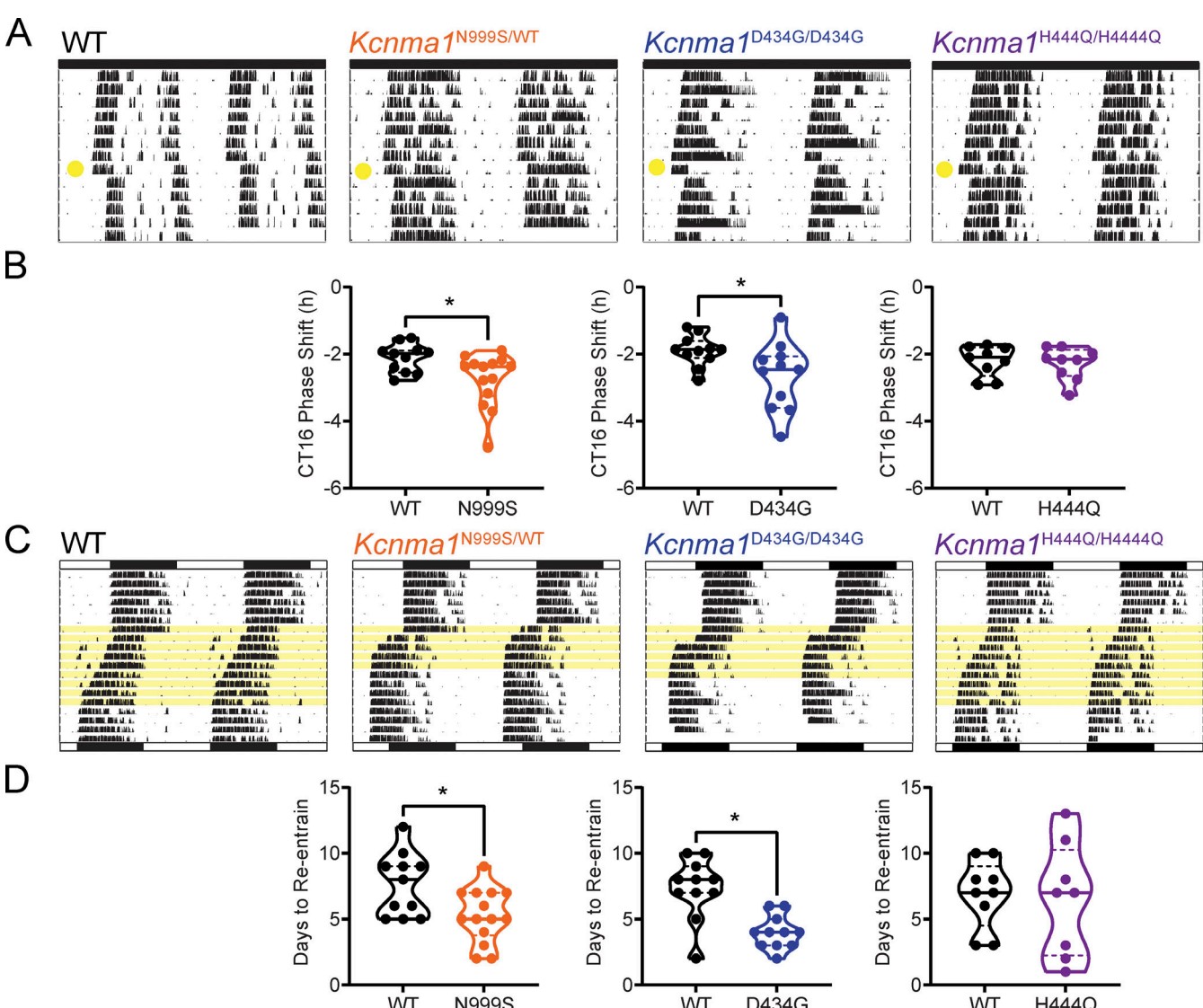

Figure 3. **Analysis of responses to phase-shifting light stimuli. (A)** Representative double-plotted running wheel actograms for WT, *Kcnma1*[N999S/WT], *Kcnma1*[D434G/D434G], and *Kcnma1*[H444Q/H444Q] mice. A 30-min light pulse was delivered 4 h into the subjective night (yellow circle indicates the day the light pulse was administered). **(B)** Number of hours of delay for WT, *Kcnma1*[N999S/WT], *Kcnma1*[D434G/D434G], and *Kcnma1*[H444Q/H444Q] mice are presented as individual data points with median and interquartile range (WT versus *Kcnma1*[N999S/WT], P = 0.0398, MWU, n = 11, 12, respectively; WT versus *Kcnma1*[D434G/D434G], P = 0.0409, t test, n = 11, 11, respectively). **(C)** Representative double-plotted running wheel actograms for WT, *Kcnma1*[N999S/WT], *Kcnma1*[D434G/D434G], and *Kcnma1*[H444Q/H444Q] mice. Mice were entrained to a LD cycle until 7 d of stable re-entrainment (indicated by bars at the top). The LD cycle was then advanced 6 h as indicated by the bar at the bottom. Yellow shaded area denotes the number of days for each mouse to re-entrain to the new LD cycle. **(D)** Number of days to stable re-entrainment for WT, *Kcnma1*[N999S/WT], *Kcnma1*[D434G/D434G], and *Kcnma1*[H444Q/H444Q] mice are presented as individual data points with median and interquartile range (WT versus *Kcnma1*[N999S/WT], P = 0.0171, t test, n = 11, 14, respectively; WT versus *Kcnma1*[D434G/D434G], P = 0.0006, t test, n = 11, 11, respectively).

*Kcnma1*[N999S/WT] and *Kcnma1*[D434G/D434G] were reduced compared with WT littermates due to a decrease in α counts (Fig. 2, A and B; and Table S1). The number of wheel rotations during the active phase decreased in *Kcnma1*[N999S/WT] mice (14,667 ± 1,432, n = 14) compared with WT (25,094 ± 1,848, n = 12; P < 0.0001, t test), and in *Kcnma1*[D434G/D434G] mice (10,989 ± 1,106, n = 11) compared with WT (24,622 ± 2,491, b = 11; P = 0.0002, t test), whereas counts during the inactive period (ρ counts) remained unchanged for both *Kcnma1*[N999S/WT] and *Kcnma1*[D434G/D434G] mice. *Kcnma1*[H444Q/H444Q] mice observed no locomotor dysfunction as evaluated by total, α or ρ counts, consistent with the lack of paroxysmal dyskinesia phenotype seen in this mouse line.

To understand the structure of locomotor running, we further assessed the consolidated activity bouts over the recording period. *Kcnma1*[N999S/WT] and *Kcnma1*[D434G/D434G] mice had the same number of bouts of the same length per day as WT littermates (Fig. 2, C and D; and Table S1). However, the number of counts during these bouts was reduced for *Kcnma1*[N999S/WT] and *Kcnma1*[D434G/D434G] mice compared with their respective WT littermates (Fig. 2 E and Table S1). *Kcnma1*[H444Q/H444Q] mice showed an increase in the number of bouts per day, consistent with the provoked hyperactive dyskinesia previously reported for this line (Park et al., 2022), but there was no change in other bout parameters assessed (Fig. 2, C–E; and Table S1). Taken together,

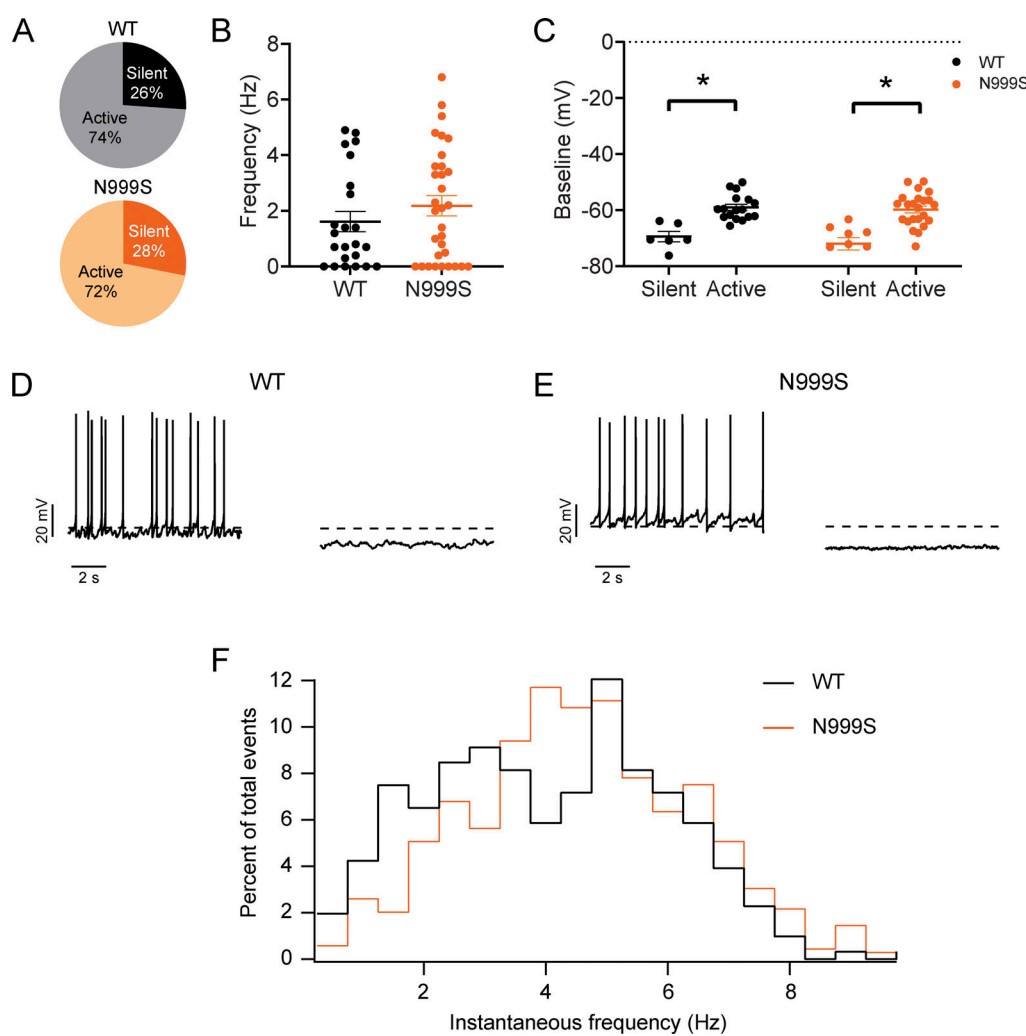

Figure 4. **Evaluation of action potential firing in *Kcnma1*[N999S/WT] SCN neurons. (A)** Relative proportion of WT and *Kcnma1*[N999S/WT] neurons that were silent and active. **(B)** Firing rate of all neurons was recorded, including silent cells. Data presented as individual values with mean ± SEM. **(C)** Baseline membrane potential for silent and active neurons. Data presented as individual values with mean ± SEM (WT silent versus active neurons, P = 0.0010, *t* test, *n* = 6 neurons, 17 neurons from 12 slices, respectively; *Kcnma1*[N999S/WT] silent versus active neurons, P = 0.0003, *t* test, *n* = 9, 23 neurons from 15 slices, respectively). **(D and E)** Representative spontaneous action potentials from WT and *Kcnma1*[N999S/WT] silent and active cells, as indicated, dotted line is −60 mV. **(F)** Spontaneous action potential frequency distribution. The histogram shows the percent of total events at each instantaneous frequency for WT (black) and *Kcnma1*[N999S/WT] (orange).

these results demonstrate that *Kcnma1*[N999S/WT] and *Kcnma1*[D434G/D434G] mice display a locomotor deficit consisting of hypoactivity, while retaining normal bout structure and duration. In contrast, *Kcnma1*[H444Q/H444Q] mice show hyperactivity within bouts, while also retaining normal bout structure.

## Evaluation of plasticity of *Kcnma1*[N999S/WT], *Kcnma1*[D434G/D434G], and *Kcnma1*[H444Q/H444Q] mice

Previous GOF models have demonstrated an increased sensitivity to phase-shifting stimuli, which results from diminished clock function (Montgomery et al, 2013; Whitt et al, 2016). Therefore, we predict that *Kcnma1*[N999S/WT] and *Kcnma1*[D434G/D434G] mice would also have an increased sensitivity phase-shifting stimuli. To test this, a 30-min light pulse was delivered during early subjective night (CT16). At CT16, the circadian system demonstrates a maximal phase delay response to light stimulation.

*Kcnma1*[N999S/WT] and *Kcnma1*[D434G/D434G] had an exaggerated response to the light pulse, 38 and 46 min longer, respectively, than WT littermates (Fig. 3, A and B; and Table S1). However, *Kcnma1*[H444Q/H444Q] had no change in the response to a pulse of light (Fig. 3, A and B; and Table S1).

The LD cycle was next advanced by 6 h to test the ability of the mice to re-entrain after a phase shift. *Kcnma1*[N999S/WT] and *Kcnma1*[D434G/D434G] mice re-entrain to the new LD cycle faster than WT littermates, ~2 and 3 fewer days, respectively (Fig. 3, C and D; and Table S1). On the other hand, *Kcnma1*[H444Q/H444Q] mice had no change in the number of days to re-entrain with the same paradigm (Fig. 3, C and D; and Table S1). However, the variability in the number of days for *Kcnma1*[H444Q/H444Q] mice to re-entrain was much larger, ranging from 1–13 d, compared with the other lines. Both results testing behavioral responses to phase-shifting stimuli suggest an increased sensitivity of *Kcnma1*[N999S/WT] and

$Kcnma1^{D434G/D434G}$ SCNs, with no differences found in $Kcnma1^{H444Q/H444Q}$.

### Evaluation of $Kcnma1^{N999S/WT}$ SCN neuronal excitability

Previous GOF models have shown a decrease in the daytime firing frequency of SCN neurons associated with behavioral changes. In these models, membranes are hyperpolarized and the proportion of silent and slowly firing neurons increases in the SCN (Montgomery et al, 2013; Whitt et al, 2016). Therefore, we hypothesize in a similar way that the GOF nature of N999S and D434G would increase BK channel current in SCN neurons, in turn decreasing spontaneous firing frequency. To test this, we used a whole-cell patch clamp in neurons of acute SCN slices to record spontaneous action potential firing and membrane properties at ZT4–8, the peak of the daily firing frequency (Fig. 4, A–F). We focused on $Kcnma1^{N999S/WT}$ mice due to N999S causing the most severe GOF effect on BK currents (Moldenhauer et al, 2020; Park et al, 2022). Our results show the same proportion of active and silent cells within SCN slices from WT (74%) and $Kcnma1^{N999S/WT}$ (72%) mice, respectively (Fig. 4 A). The average spontaneous firing frequency of active neurons was also unchanged in $Kcnma1^{N999S/WT}$ 3.0 ± 0.4 Hz ($n$ = 23 neurons, 15 slices) when compared with WT littermates (2.2 ± 0.4 Hz, $n$ = 17 neurons, 12 slices; Fig. 4 B).

Other neuronal parameters were unchanged as well. The baseline membrane potential of $Kcnma1^{N999S/WT}$ active and silent cells was not different from WT (Fig. 4 C). However, the baseline membrane potential of silent cells was hyperpolarized compared with active cells within each genotype, as expected (Fig. 4, C and D). Additional membrane parameters were also similar. Input resistance in $Kcnma1^{N999S/WT}$ neurons was not significantly different from WT (2.7 ± 0.3 GΩ, $n$ = 22 neurons, 10 slices, versus 2.3 ± 0.2 GΩ, $n$ = 24 neurons, 11 slices, respectively; $P$ = 0.33, $t$ test). Capacitance of $Kcnma1^{N999S/WT}$ neurons was not significantly different from WT (12.7 ± 0.6 pF, $n$ = 23, 12 slices versus 11.8 ± 0.5 pF, $n$ = 32, 15 slices, respectively; $P$ = 0.30, $t$ test). Lastly, a more detailed cumulative distribution analysis of the instantaneous firing rates did not reveal a statistically significant difference in firing patterns (Fig. 4 F).

However, analysis of individual action potential waveforms between 3 and 4.5 Hz revealed changes in the baseline membrane potential ($Kcnma1^{N999S/WT}$: −57.5 ± 0.4 mV, $n$ = 195 events from 15 cells, and WT: −53.8 ± 0.5 mV, $n$ = 72 events from 5 cells; $P$ < 0.0001, MWU), peak potential ($Kcnma1^{N999S/WT}$: 49.4 ± 0.8 mV, $n$ = 195 events from 15 cells, and WT: 41.4 ± 1.2 mV, $n$ = 72 events from 5 cells; $P$ < 0.0001, MWU), and after hyperpolarization potential ($Kcnma1^{N999S/WT}$: −62.7 ± 0.5 mV, $n$ = 195 events from 15 cells, and WT: −58.5 ± 0.5 mV, $N$ = 72 events from 5 cells; $P$ = 0.0056, MWU). Hyperpolarization of the baseline potential and AHP is consistent with increased BK channel activity observed in other transgenic manipulations or other neuron types (Montgomery et al, 2013; Whitt et al, 2018; Park et al, 2022). Thus, although there were no overall changes in the net firing output under spontaneous firing conditions with intact synaptic transmission, changes in action potential waveforms suggest the N999S GOF effect can be detected in these excitability parameters from SCN neurons.

## Discussion

In this study, to test the hypothesis that human patient mutations in the $KCNMA1$ gene would cause a disruption in the circadian rhythm, we assessed the circadian behavior for three $KCNMA1$ channelopathy mouse models. We identified limited evidence for a primary circadian deficit in the two GOF mouse lines evaluated, with no effect in the LOF mice. $Kcnma1^{D434G/D434G}$ showed a small (13 min) reduction in average period and a significant (23–34%) reduction in circadian amplitude. $Kcnma1^{N999S/WT}$ showed an 18–35% reduction only in circadian amplitude, with wider variability in the period compared with WT that precluded identifying a similar change in period (Fig. 1). The reduction in circadian amplitude could be part of a primary circadian pacemaker deficit, as signaled by the small change in period, or could result from the dyskinesia-related locomotor dysfunction (Dong et al, 2022; Park et al, 2022). Consistent with this, $Kcnma1^{D434G/D434G}$ and $Kcnma1^{N999S/WT}$ mice showed large reductions in wheel running compared with their WT littermates, 52% and 40% less, respectively, without alterations in the timing of the activity (α or bout lengths; Fig. 1 F and Fig. 2). Since decreased locomotor activity through other mechanisms such as restricting wheel access has been shown to lengthen, not shorten, the circadian period (Edgar et al, 1991), it suggests that hypoactivity in $Kcnma1^{D434G/D434G}$ is not the main contributor to the change in period. Nevertheless, a definitive delineation of the primary circadian deficit without the current caveat of hypoactivity will require additional investigations employing home cage activity, where locomotor dyskinesia may not be similarly provoked. Conditional mutations could also be employed to investigate specific GOF effects on the SCN without dyskinesia motor complications; however, the transgenic mouse lines used in this study are not available as conditional mutations (Park et al, 2022).

While these data suggest the SCN clock in both $Kcnma1^{D434G/D434G}$ and $Kcnma1^{N999S/WT}$ channelopathy models can function to appropriate circadian behavioral activity to the correct timing and phase, some clear deficits were evident in clock-controlled responses to zeitgebers that are associated with low-amplitude circadian oscillators. $Kcnma1^{D434G/D434G}$ and $Kcnma1^{N999S/WT}$ mice demonstrated increased sensitivity to phase-shifting light stimuli (Fig. 3), consistent with a decrease in SCN function. However, BK channels in the retina (Grimes et al, 2009) regulating light input to the SCN could also affect phase shifting. We favor the SCN interpretation because all transgenic lines entrain normally to a standard LD cycle (Table S1), although future experiments could test the function of the retina directly, as well as other phase-shifting paradigms. The phenotypes in these two mouse lines extend previous $Kcnma1$-related findings to demonstrate that five separate BK channel manipulations impact clock function. In particular, there is a new consensus highlighted by the commonality among four GOF mouse lines of increased sensitivity to phase-shifting stimuli (Meredith et al, 2006; Montgomery et al, 2013). These results and prior studies generate a qualitative allelic series in the severity of circadian rhythm disruption of $Kcnma1^{-/-}$ > β2 KO > Tg-BK$^{R207Q}$ > $Kcnma1^{D434G/D434G}$ > $Kcnma1^{N999S/WT}$ >> $Kcnma1^{H444Q/H444Q}$. However, a direct comparison of the GOF models in this study with

the previous GOF *Tg*-BK$^{R207Q}$ and β2 KO models experiments would require a parallel assessment under conditions where the output of the circadian rhythm is not dependent on the physical ability of the mice.

In the GOF mouse lines *Tg*-BK$^{R207Q}$ and β2 KO mice, the increased response to a CT16 light pulse and the ability to more quickly re-entrain to phase advance of the LD cycle occurred in the setting of altered daytime SCN firing rates (Montgomery and Meredith, 2012; Montgomery et al, 2013; Whitt et al, 2016). Yet *Kcnma1*$^{N999S/WT}$ SCN neurons did not show significant differences in firing rate, despite the changes in action potential waveform observed in the midrange of the firing activity distribution that was consistent with alterations previously observed in dentate granule neurons of the hippocampus (Park et al., 2022). While the hyperpolarization of the baseline and AHP was indicative of the increased BK channel activity in *Kcnma1*$^{N999S/WT}$ neurons, it is clear that these differences in the spontaneous firing curve were not sufficient to alter the circadian behavior of *Kcnma1*$^{N999S/WT}$ mice. In other studies using GOF transgenic mouse lines, decreases of about 30% (*Tg*-BK$^{R207Q}$) and ~75% (β2 KO) in the daytime SCN firing are associated with changes in the circadian behavior (Montgomery et al., 2013; Whitt et al., 2016). Several factors may contribute to the lack of net firing difference in this study, including that the electrophysiology was conducted at only a single time point (daytime) compared with previous multi-timepoint studies. Greater differences may be revealed if the GOF effects for the mutations could be assessed at times of day when the fast-delayed rectifier K$^+$ channel, which plays a large role in setting daytime firing rates by promoting rapid repolarization, is not dominant (Itri et al., 2005; Montgomery and Meredith, 2012). In addition, restricting recordings to a specific neuronal subpopulation may be necessary to detect changes in firing. For example, alterations in the speed of behavioral rhythm entrainment differed in VIP-expressing neurons with neuronal frequency (Mazuski et al., 2018). Thus, while the lack of change at the ZT4-8 peak SCN firing is consistent overall with the lack of a strong effect on primary circadian rhythm parameters in *Kcnma1*$^{N999S/WT}$ mice, it remains to be determined what the SCN-level defect is related to the increased phase-shifting behaviors.

In comparison with other GOF models previously studied, the extent of the primary circadian behavioral deficits is variable (Montgomery et al., 2013; Whitt et al., 2016). *Tg*-BK$^{R207Q}$ showed an elongation of the active period (Montgomery et al., 2013), which is not seen in other GOF mice models (*Kcnma1*$^{N999S/WT}$, *Kcnma1*$^{D434G/D434G}$, or β2 KO), where the active period is comparable between mutant and WT mice. Like the GOF mouse lines in this study, β2 KO mice also showed a reduction in the circadian amplitude suggesting a diminished clock. However, these mice increased their wheel activity during the inactive period (ρ counts), demonstrating that the lack of β2 hampers the ability of the mice to distinguish the active and inactive portions of the cycle (Whitt et al., 2016). This disruption was not identified in *Tg*-BK$^{R207Q}$, *Kcnma1*$^{N999S/WT}$, and *Kcnma1*$^{D434G/D434G}$ mice. However, none of these studies separate the behavioral analysis by sex, and it is possible that this could provide additional insights that aid direct comparisons between these lines.

From this study, *KCNMA1* channelopathy patients harboring N999S or D434G mutations might not be expected to have a primary circadian deficit that contributes to the anecdotal reports of sleep symptoms. However, the phenotype of *Kcnma1*$^{N999S/WT}$ and *Kcnma1*$^{D434G/D434G}$ mice predict these patients could be more sensitive to jet lag or other circadian rhythm disturbances. Although patients circumstantially report sleep disturbance and high sensitivity to sleep loss, this has not been directly addressed yet in *KCNMA1* channelopathy patient clinical genotype–phenotype correlations (Bailey et al, 2019; Miller et al, 2021). Uncovering the basis for patient sleep disturbances remains an interesting new dimension within the neurological landscape of *KCNMA1* channelopathy.

## Data avaliability

The data for all figures and supplementary table are openly available in DRYAD at https://doi.org/10.5061/dryad.3bk3j9kr0. ClockLab files are available from the corresponding author upon reasonable request.

## Acknowledgments

Jeanne M. Nerbonne served as editor.

We thank Kelly Tammen and Indira Jetton for assistance with animals and Hans Moldenhauer for comments on the manuscript.

This work was supported by National Heart, Lung, and Blood Institute grant R01-HL102758 (to A.L. Meredith) and the Training Program in Integrative Membrane Biology National Institute of General Medical Sciences grant T32-GM008181 (to A.L. Meredith).

Author contributions: R.L. Dinsdale: research conception and data collection, data analysis, statistical analysis design and execution, and manuscript writing. C.E. Roache: research conception and data collection, data analysis, statistical analysis design, and execution. A.L. Meredith: research conception, data analysis, and manuscript writing. All authors approved the final version of the manuscript.

Disclosures: The authors declare no competing interests exist.

Submitted: 30 January 2023

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

# Supplemental material

One table is provided online. Table S1 provides a summary of behavioral data used in this study.

