## [Peer Review File · The Journal of General Physiology]

Disease-associated KCNMA1 variants decrease circadian clock robustness in channelopathy mouse models

Ria Dinsdale, Cooper Roache, and Andrea Meredith

Corresponding Author(s): Andrea Meredith, University of Maryland School of Medicine

Review Timeline:

Submission Date:	January 30, 2023
Editorial Decision:	March 15, 2023
Revision Received:	July 7, 2023
Editorial Decision:	July 25, 2023
Revision Received:	August 24, 2023

Editor: Jeanne Nerbonne

Transaction Report:

DOI: <https://doi.org/10.1085/jgp.202313357>

March 16, 2023

Dr. Andrea L Meredith
University of Maryland School of Medicine
Dept. of Physiology
655 W. Baltimore St.
BRB 5-029
Baltimore, MD 21201

Re: 202313357

Dear Andrea,

Thank you for submitting your manuscript, titled "Evaluation of circadian behavior in KCNMA1 channelopathy mouse models" to JGP. Your manuscript has now been seen by 3 reviewers, whose comments are appended below. As you will see, the reviewers have raised several concerns about the descriptions of the animals used in the experiments, as well as about some of the specific experiments and analyses completed and your conclusions about how the results presented in this manuscript (do or do not) suggest a major role for KCNMA1-encoded channels in the regulation of circadian rhythms in the firing properties of SCN neurons and/or wheel running (locomotor) activity (in mice). The editors concur with these concerns. In addition, the editors noted that, throughout the text, the different mouse lines used are referred to as different strains; but it seems likely (based on the descriptions of the mice provided in the 2022 eLife paper from this groups that all lines are the same (C57Bl6J) strain.

If you can address the concerns identified, we would be happy pleased to receive a revised version of this manuscript for further consideration. Please be aware that we all have your revised submission re-reviewed, preferably by the original referees, pending their availability. Based on the scope of the requested changes, we would expect that the revision process will take no longer than 3-4 months. If, however, you need additional time, please do let us know. In addition, please do not hesitate to contact me if you feel that a discussion of the reviewers' and editors' comments would be helpful.

Please submit your revised manuscript via the link below along with a point-by-point letter that details your responses to the editors' and reviewers' comments, as well as a copy of the text with alterations highlighted (boldfaced or underlined). If the article is eventually accepted, it would include a 'revised date' as well as submitted and accepted dates. If we do not receive the revised manuscript within one year, we will regard the article as having been withdrawn. We would be willing to receive a revision of the manuscript at a later time, but the manuscript will then be treated as a new submission, with a new manuscript number.

Please pay particular attention to recent changes to our instructions to authors in sections: Data presentation, Blinding and randomization and Statistical analysis, under Materials and Methods, as shown here: <https://rupress.org/jgp/pages/submission-guidelines#prepare>. Re-review will be contingent on inclusion of the required information (including for data added during revision) and demonstration of the experimental reproducibility of the results (i.e., all experimental data verified in at least 2 independent experiments).

When revising your manuscript, please be sure it is a double-spaced MS Word file and that it includes editable tables, if appropriate.

Please submit your revised manuscript via this link:
Link Not Available

Thank you for the opportunity to consider your manuscript.

Sincerely,

Jeanne

Jeanne M. Nerbonne, Ph.D.
On behalf of Journal of General Physiology

Journal of General Physiology's mission is to publish mechanistic and quantitative molecular and cellular physiology of the highest quality; to provide a best-in-class author experience; and to nurture future generations of independent researchers.

The authors have characterized the circadian response and behavioral activity of three (2 GOF and 1 LOF) KCNMA1 mutations (of human KCNAM1 channelopathies) in transgenic mice. The authors demonstrated that phase shifting sensitivity was abnormal, there were delayed increased responses to light pulses and took fewer days to re-entrain to a new light:dark cycle.

The authors have done a good in advancing their research of these GOF and LOF KCNMA1 mutants to understand the effects on circadian clock behaviors in their transgenic mice. The data is well presented and the manuscript is well written. A few minor comments follow.

1) Why was the KCNMA1 N999S/WT mouse used as the WT control instead of true WT mice for the actograms of all figures?

2) The authors did not find any significant changes in net firing of action potentials between the WT and N999S/WT brain slices, they however saw some individual differences in AP waveforms between 3 and 4.5 Hz revealing changes in baseline MP and afterhyperpolarization. While they mentioned this is consistent with GOF BK channel activity, a further explanation of the physiological relevance of these differences particularly with respect to the circadian behaviors they observe in their study would be beneficial.

Reviewer #2 (Comments to the Authors):

KCNMA1 channels play a commanding role in the regulation of circadian behaviors. Here, Dinsdale and colleagues explored whether three KCNMA1 point mutations found in humans, alter circadian behavior in mice. Three mutations were studied: two of which (N999S and D434G) increased KCNMA1 activity (gain-of-function), whereas the third H444Q reduced channel activity (loss-of-function). All three mutants have been validated as models of paroxysmal dyskinesia. Circadian rhythms were assessed by activity while mice were on established light:dark cycles compared to dark only conditions.

The findings demonstrate that gain-of-function mutations had normal circadian behavioral activity, with some deficits to phase shifting stimuli.

My only minor feedback is as a reader with channel expertise, I was able to understand all of the analysis and how they related to interpretations regarding circadian rhythms - with the exception of the fast fourier transforms. It might be useful to provide a sentence relating how this particular data relates to the circadian rhythm behavior.

Overall, the experiments are well described, properly controlled, and reveal compelling data about how KCNMA1 channels contribute to circadian behaviors. This paper was well written and I support the acceptance of this short report.

Reviewer #3 (Comments to the Authors):

In this manuscript, Dinsdale and colleagues report the outcome of studies investigating gain of function and loss of function mutations in *Kcnma1*, which codes for the BK channel. The Meredith lab have shown that the BK channel contribution to the electrophysiological activity of the suprachiasmatic nuclei (SCN), varies over the day and night. This is not unexpected as the SCN contain the dominant circadian pacemaker in the mammalian brain. The Meredith lab have done a lot of research in this area and teasing apart the various contributions of B₁ and the effects of BK mutants on SCN neurophysiology as well as circadian rhythms in behavior. The authors report that there are subtle differences in the circadian behavior of the mice lines, with a reduction in the amount of locomotor activity over the circadian cycle and shifting to acute and long-term changes in the lighting conditions being the more robust. In the mouse line with the larger reported effects on BK current (*Kcnma1*N999s/wt), the authors evaluated parameters of SCN neuronal activity over the middle of the day (ZT4-8). They find that in comparison to recordings from the SCN of WT littermates, most parameters are unchanged. One difference is that for SCN neurons firing over 3-4.5 spikes/s, the neurons are more hyperpolarized in the *Kcnma1*N999s/wt SCN cells, while differences were also seen in peak potential and afterhyperpolarization potential. Overall, the study seems to have been conducted to a good standard, however, there are some points for the authors to consider.

1) It would be useful to have a proper description of the genotypes of the mice--are all the backgrounds fully C57Bl6 or are there differences? What was the ratio of male to female mice used in the different lines--are they similar?

2) It is not clear why a limited time of day was used to sample the SCN neurons---since there is a reduction in locomotor activity in the mutants, why not sample from the night-time to see if SCN activity is unusual then?

3) The phase-resetting to light pulses or an advance in the LD cycle yielded some large differences. However, as the SCN does not appear to be grossly affected by LOF or GOF mutations in BK, then other sites of action of light need to be considered. For example, *Kcnma1* is expressed in the retina--perhaps these mutations affect communication from the retina to the SCN or indeed the affect how the retina processes light information. These possibilities should be tested.

4) Related to 3), the behavioral analysis would be boosted by the inclusion of mice that receive the advance in the LD cycle for 48h and are then transferred into constant dark. This would allow quantification of the extent to which they reset over the initial 48h.

We thank the editor and reviewers for the comments on this Brief Communication submission to JGP. The main changes to the manuscript are text additions. Per the editor and reviewer comments, we clarify the descriptions of the mouse lines used in the experiments by adding an additional paragraph in the methods section (lines 132-143). We reviewed the manuscript to ensure consistent use of mouse “lines” throughout.

We have also added additional text details within the results and discussion to add clarity to the manuscript as suggested by the specific reviewer points below.

Reviewer #1 (Comments to the Authors):

The authors have characterized the circadian response and behavioral activity of three (2 GOF and 1 LOF) KCNMA1 mutations (of human KCNAM1 channelopathies) in transgenic mice. The authors demonstrated that phase shifting sensitivity was abnormal, there were delayed increased responses to light pulses and took fewer days to re-entrain to a new light:dark cycle.

The authors have done a good in advancing their research of these GOF and LOF KCNMA1 mutants to understand the effects on circadian clock behaviors in their transgenic mice. The data is well presented, and the manuscript is well written. A few minor comments follow.

1) Why was the KCNMA1 N999S/WT mouse used as the WT control instead of true WT mice for the actograms of all figures?

We used separate cohorts of WT littermates for EACH of the 3 transgenic lines in all sets of the experiments (eg in Figure 1, panels C-F). This point has been clarified by stating the breeding schemes for each line in the methods section.

As stated originally, the representative WT actogram for all the figures is a WT mouse from the *Kcnma1*^{N999S} line (eg Figure 1A-B). We now more clearly emphasize this point in the methods with the addition on lines 166-167: “For all figures, the WT control actogram is a WT mouse from a *Kcnma1*^{N999S/WT} x C57Bl/6J litter.”

Displaying only a single WT actogram was done to simplify the presentation of the representative behavioral data down to 4 panels. If we had displayed a representative actogram for EACH of the WTs for all 3 lines, it would have required a 6-panel figure for each behavioral parameter that we analysed, which did not display as well (it made the figures much too dense and the axes labels unreadable).

2) The authors did not find any significant changes in net firing of action potentials between the WT and N999S/WT brain slices, they however saw some individual differences in AP waveforms between 3 and 4.5 Hz revealing changes in baseline MP and afterhyperpolarization. While they mentioned this is consistent with GOF BK channel activity, a further explanation of the physiological relevance of these differences particularly with respect to the circadian behaviors they observe in their study would be beneficial.

When BK current increases due to GOF channel activity, the baseline membrane potential and action potential AHP can become hyperpolarized. These changes are a telltale sign of increased BK current during the action potential. We added this clarification to lines 321-322 of the Results.

We agree that, while this is likely indicative of the increased BK channel activity in *Kcnma1*^{N999S/WT} SCN neurons, it is clear that these differences in the spontaneous firing curve are not sufficient to alter the circadian behavior. In other studies using GOF transgenic mouse lines, we found that decreases in the daytime SCN firing of about 30% (*Tg-BK*^{R207Q}) and ~75 % (β 2 KO) were associated with detectable changes in circadian behavior (Montgomery et al., 2013, Whitt et al., 2016). This information has been added to the Discussion on lines 375-382.

Reviewer #2 (Comments to the Authors):

KCNMA1 channels play a commanding role in the regulation of circadian behaviors. Here, Dinsdale and colleagues explored whether their KCNMA1 point mutations found in humans, alter circadian behavior in mice. Three mutations were studied: two of which (N999S and D434G) increased KCNMA1 activity (gain-of-function), whereas the third H444Q reduced channel activity (loss-of-function). All three mutants have been validated as models of paroxysmal dyskinesia. Circadian rhythms were assessed by activity while mice were

on established light:dark cycles compared to dark only conditions. The findings demonstrate that gain-of-function mutations had normal circadian behavioral activity, with some deficits to phase shifting stimuli.

My only minor feedback is as a reader with channel expertise, I was able to understand all of the analysis and how they related to interpretations regarding circadian rhythms - with the exception of the fast fourier transforms. It might be useful to provide a sentence relating how this particular data relates to the circadian rhythm behavior.

To add more clarity for readers without expertise in circadian behavioural analysis, we have included these additional details and a new reference in the results section on lines 230-233: The paragraph now reads: "To assess circadian behavioral amplitude in DD, the circadian peak from χ^2 periodogram analysis and Fast Fourier Transforms were quantified. Both tests assess the robustness of the circadian rhythm from the amplitude of the circadian signal, using the peak of the χ^2 periodogram signal and the relative power of the circadian frequency band (0.04 to 0.042 cycles per hour), respectively (Sokolove and Bushell, 1978)."

Overall, the experiments are well described, properly controlled, and reveal compelling data about how KCNMA1 channels contribute to circadian behaviors. This paper was well written and I support the acceptance of this short report.

Reviewer #3 (Comments to the Authors):

In this manuscript, Dinsdale and colleagues report the outcome of studies investigating gain of function and loss of function mutations in *Kcnma1*, which codes for the BK channel. The Meredith lab have shown that the BK channel contribution to the electrophysiological activity of the suprachiasmatic nuclei (SCN), varies over the day and night. This is not unexpected as the SCN contain the dominant circadian pacemaker in the mammalian brain. The Meredith lab have done a lot of research in this area and teasing apart the various contributions of BK and the effects of BK mutants on SCN neurophysiology as well as circadian rhythms in behavior. The authors report that there are subtle differences in the circadian behavior of the mice lines, with a reduction in the amount of locomotor activity over the circadian cycle and shifting to acute and long-term changes in the lighting conditions being the more robust. In the mouse line with the larger reported effects on BK current (*Kcnma1*N999s/wt), the authors evaluated parameters of SCN neuronal activity over the middle of the day (ZT4-8). They find that in comparison to recordings from the SCN of WT littermates, most parameters are unchanged. One difference is that for SCN neurons firing over 3-4.5 spikes/s, the neurons are more hyperpolarized in the *Kcnma1*N999s/wt SCN cells, while differences were also seen in peak potential and afterhyperpolarization potential. Overall, the study seems to have been conducted to a good standard, however, there are some points for the authors to consider.

1) It would be useful to have a proper description of the genotypes of the mice--are all the backgrounds fully C57BL6 or are there differences? What was the ratio of male to female mice used in the different lines--are they similar?

All mice are on a C57BL6/J background (added to the methods at line 133-134). The parent breeding crosses that produced the transgenics and WT littermates, as well as the genotyping information, has been clarified (added at lines 134-138). The number of females and males for all experiments has also been added to the Supplemental Table 1. A sentence was added to the methods indicating that the data were obtained from both sexes with no randomization (line 164).

Given the difficulty of generating the transgenic genotypes and colony limitations during the pandemic, these are the only transgenic mice produced that we could perform the experiments with. We did perform a power calculation to see if the existing dataset could be used for a sex-delineated analysis. Not surprisingly, this analysis indicated that separating the data by sex was too underpowered. Thus, no conclusions could be made from the existing data that would merit including in the manuscript. Addressing sex as a biological variable in the behavioral parameters will require a much larger study and is beyond the scope of our current capabilities. This acknowledgement has been added to lines 402-403 of the discussion, where the comparison of phenotypes between the 4 GOF transgenic lines (2 in this study, and the 2 published

previously) are discussed. It is possible that designing a larger cohort study, that is powered to detect sex-based differences, could provide deeper insights into the similarities demonstrated between the lines.

2) It is not clear why a limited time of day was used to sample the SCN neurons---since there is a reduction in locomotor activity in the mutants, why not sample from the night-time to see if SCN activity is unusual then?

We recorded spontaneous firing during the daytime based on clear predictions from previously generated BK channel GOF transgenic mouse lines (*Tg-BK^{R207Q}* and $\beta 2$ KO). *Kcnma1^{N999S/WT}* (and *Kcnma1^{D434G/D434G}*) mice have circadian behavioral alterations that are similar to *Tg-BK^{R207Q}* or $\beta 2$ KO (these points are already discussed in detail in lines 359 and onwards in the Discussion). One commonality between these previously generated GOF lines is the strong reduction in daytime firing (*Tg-BK^{R207Q}* has a 30% decrease in firing; $\beta 2$ KO has a ~75 % decrease)(Montgomery et al., 2013, Whitt et al., 2016). In contrast, there was no major difference in night-time firing rates in either mouse line. Therefore, we predicted that the daytime firing might have a similar alteration in *Kcnma1^{N999S/WT}* neurons (stated in lines 292-293). From these previous studies, the nighttime point was not predicted to alter firing as strongly, if at all.

Moreover, on a practical note, these experiments were conducted during the pandemic and its aftermath. We were only able to test a single mouse line due to ongoing staffing issues in the lab and animal facility, which caps our colony size. Nighttime experiments require extra monitoring to maintain animals on a reverse light-dark cycle in satellite housing (seven days a week, falling under the lab's responsibility). After discussing a variety of options, we conclude that we simply do not currently have the staffing and resources to conduct nighttime recordings for this study. Therefore, in this Brief Communication submission, we are able to answer the first question of whether the daytime firing is altered, which is the single timepoint with the most evidence and background rationale suggesting the potential to differ. The limitations were acknowledged already in the discussion section, lines 381-392.

3) The phase-resetting to light pulses or an advance in the LD cycle yielded some large differences. However, as the SCN does not appear to be grossly affected by LOF or GOF mutations in BK, then other sites of action of light need to be considered. For example, *Kcnma1* is expressed in the retina--perhaps these mutations affect communication from the retina to the SCN or indeed the affect how the retina processes light information. These possibilities should be tested.

The role, if any, for BK channels in the retina regulating circadian rhythm has not been studied. If the mutations in *Kcnma1* affected light processing by the retina, is it possible we would have detected this in the entrainment to the standard 12:12-h light:dark (LD) cycle. However, the parameters analysed from LD (circadian period, χ^2 amplitude and FFT rPSD) were not different between any of the transgenic mice and WT littermates (results section, lines 213-216).

We cannot conduct additional experiments to test this directly, with the same limitations as discussed above in response #2. Instead, we have added a sentence in the discussion to state the potential effect of the mutations on light processing by the retina, "However, BK channels in the retina (Grimes et al., 2009) regulating light input to the SCN could also affect phase shifting. We favour the SCN interpretation because all transgenic lines entrain normally to a standard light:dark cycle (Supplementary Table 1), although future experiments could test the function of the retina directly, as well as other phase-shifting paradigms." lines 355-359.

4) Related to 3), the behavioral analysis would be boosted by the inclusion of mice that receive the advance in the LD cycle for 48h and are then transferred into constant dark. This would allow quantification of the extent to which they reset over the initial 48h.

This experiment is a good suggestion for further investigation into the mechanism of the phase-shifting alterations and autonomous SCN function. However, it is beyond the scope of the current short report, which focuses on the two light stimulus paradigms used in our prior studies. The second part of the sentence added in response to the prior comment addresses now addresses this by stating, 'future experiments could test.....other phase-shifting paradigms.'

July 26, 2023

Dr. Andrea L Meredith
University of Maryland School of Medicine
Dept. of Physiology
655 W. Baltimore St.
BRB 5-029
Baltimore, MD 21201

Re: 202313357R1

Dear Andrea,

I am pleased to let you know that your manuscript, titled "Evaluation of circadian behavior in KCNMA1 channelopathy mouse models", is scientifically acceptable for publication in Journal of General Physiology. Formal acceptance will follow when it is modified in accordance with our editorial policies.

Please note items that need attention are listed at the bottom of this email (under 'manuscript formatting checklist') and on the attached marked-up pdf file. Please also be sure to include a copy of the text with alterations highlighted (boldfaced or underlined). Your manuscript should be a double-spaced MS Word file and include editable tables, if appropriate.

Please also note that JGP now requires a data availability statement for all research article submissions. These statements will be published in the article directly above the Acknowledgments. The statement should address all data underlying the research presented in the manuscript. Please visit the JGP instructions for authors for guidelines and examples of statements at <https://rupress.org/jgp/pages/editorial-policies#data-availability-statement>.

Please submit your final files via this link:
Link Not Available

Thank you for choosing to publish your research in JGP and please feel free to contact me with any questions.

Sincerely,

Jeanne

Jeanne Nerbonne, Ph.D.
On behalf of Journal of General Physiology

Journal of General Physiology's mission is to publish mechanistic and quantitative molecular and cellular physiology of the highest quality; to provide a best in class author experience; and to nurture future generations of independent researchers.

Manuscript formatting checklist:

- MS Word document of text needed (including editable tables)
- MS Word document of supplemental text needed, if applicable (including figure legends and editable tables)
- Brief Statement describing supplementary information needed, if applicable (in subsection at end of Materials & Methods)
- Please include a data availability statement preceding the Acknowledgments section. Please see <https://rupress.org/jgp/pages/editorial-policies#data-availability-statement>
- Figures created at sufficient resolution and in acceptable format (including supplemental if applicable). If working in Illustrator, we prefer .ai or .eps file format. If working in Photoshop please use 600dpi/1000dpi .tiff or .psd file format. Minimum resolution at estimated print size: Minimum resolution for all figures is 600 dpi. For figures that contain both photographs and line art or text, 600 dpi is highly recommended. Figures containing only black and white elements (line art, no color, and no gray) should be 1,000 dpi. Maximum figure size is 7 in wide x 9 in high (17.5 x 22.8 cm) at the correct resolution. <https://jgp.rupress.org/fig-vid-guidelines>
- Supplemental figures, if any, conforming to same guidelines as manuscript figures (noted above)
- If images resemble one from a prior publications, the author must seek permissions (to reproduce or adapt) from the original publisher. [You can resubmit your paper while waiting to hear back from the original publisher but please keep us updated]
- All authors must complete a disclosure form prior to acceptance. A link to complete the form has been sent to all coauthors. Please provide the editorial office with updated email addresses if necessary

Reviewer #1 (Comments to the Authors):

The authors have sufficiently addressed all my questions. I have no further comments for the authors to address for their manuscript.

Reviewer #2 (Comments to the Authors):

The authors have addressed my concern. I believe that this manuscript is ready for publication.

Reviewer #3 (Comments to the Authors):

The authors have done a good job responding to the concerns raised in the previous round of reviews and I have no further recommended changes etc.